# Identification of fidelity-governing factors in human recombinases DMC1 and RAD51 from cryo-EM structures

Shih-Chi Luo[1,4], Hsin-Yi Yeh[2,4], Wei-Hsuan Lan[3], Yi-Min Wu[1], Cheng-Han Yang [1], Hao-Yen Chang[2], Guan-Chin Su[2], Chia-Yi Lee[2], Wen-Jin Wu [1], Hung-Wen Li [3], Meng-Chiao Ho [1,2✉], Peter Chi [1,2✉] & Ming-Daw Tsai [1,2✉]

Both high-fidelity and mismatch-tolerant recombination, catalyzed by RAD51 and DMC1 recombinases, respectively, are indispensable for genomic integrity. Here, we use cryo-EM, MD simulation and functional analysis to elucidate the structural basis for the mismatch tolerance of DMC1. Structural analysis of DMC1 presynaptic and postsynaptic complexes suggested that the lineage-specific Loop 1 Gln244 (Met243 in RAD51) may help stabilize DNA backbone, whereas Loop 2 Pro274 and Gly275 (Val273/Asp274 in RAD51) may provide an open "triplet gate" for mismatch tolerance. In support, DMC1-Q244M displayed marked increase in DNA dynamics, leading to unobservable DNA map. MD simulation showed highly dispersive mismatched DNA ensemble in RAD51 but well-converged DNA in DMC1 and RAD51-V273P/D274G. Replacing Loop 1 or Loop 2 residues in DMC1 with RAD51 counterparts enhanced DMC1 fidelity, while reciprocal mutations in RAD51 attenuated its fidelity. Our results show that three Loop 1/Loop 2 residues jointly enact contrasting fidelities of DNA recombinases.

[1] Institute of Biological Chemistry, Academia Sinica, Taipei, Taiwan. [2] Institute of Biochemical Sciences, National Taiwan University, Taipei, Taiwan. [3] Department of Chemistry, National Taiwan University, Taipei, Taiwan. [4] These authors contributed equally: Shih-Chi Luo, Hsin-Yi Yeh. ✉email: joeho@gate. sinica.edu.tw; peterhchi@ntu.edu.tw; mdtsai@gate.sinica.edu.tw

Homologous recombination (HR) is a well-conserved cellular process for the repair of DNA double-strand breaks[1–4], the rescue of stalled/collapsed replication forks[2,5], and meiosis[6–8]. During HR, a recombinase binds to single-stranded DNA (ssDNA) and forms a right-handed helical nucleoprotein filament, known as a presynaptic filament, which then searches double-stranded DNA (dsDNA) for homology. Upon encountering a homologous sequence, the presynaptic filament pairs with the complementary strand, resulting in the displacement of the non-complementary strand from the duplex to generate a displacement loop structure and promote DNA strand exchange[9–12].

Eukaryotic RAD51 and DMC1 both catalyze strand displacement and strand exchange. Interestingly, DMC1 possesses the ability of stabilizing mismatch-containing recombination intermediates (low-fidelity recombinase), whereas the RAD51 lacks the ability to do so (high-fidelity recombinase)[13–17]. RAD51 and DMC1 are members of the RecA family of DNA recombinases and share ~50% amino acid identity and similar biochemical properties[6,18–21]. Both are ATP-dependent DNA-binding proteins that engage DNA in nucleotide triplet clusters and extend the DNA to ~1.5-fold[18,22–25]. The bound DNA is organized into near B-form base triplets separated by ~8 Å between adjacent triplets[26]. The strand invasion products mediated by RAD51 and DMC1 filaments are both stabilized in three-nucleotide steps[27,28]. Interestingly, single-molecule studies also showed that DMC1 can stabilize single, double, or triple mismatches within a triplet in heteroduplex DNA joints, whereas RAD51 cannot[27,29,30]. It has been suggested that DNA-binding loop 1 (Loop 1) and loop 2 (Loop 2) of recombinases harbor specifically conserved amino acids within either the RAD51 lineage or the DMC1 lineage[17,26,31–34]. Recently, the cryogenic electron microscopy (cryo-EM) structures of human RAD51 complexes with an ssDNA (presynaptic) and a dsDNA (postsynaptic) were reported, which provided additional insight in the mechanism of strand exchange[28,35]. However, as there are eight lineage-specific residues in Loop 1 and Loop 2 (Fig. 1a)[17], and as no structure of DMC1-DNA filaments is available, the structural basis of the contrasting fidelity between DMC1 and RAD51 remains a major puzzle.

In this work we use cryo-EM, molecular dynamics (MD) simulations, and two independent functional assays to tackle these problems. Our results lead to a dynamics-based mechanism for recombination fidelity that involves two controlling factors: a gate for the basepair triplet and a support for the DNA backbone. A tight gate and a loose backbone support contribute to the high fidelity of RAD51, whereas a loose gate and a tight backbone support contribute to the mismatch tolerance of DMC1.

## Results

### Cryo-EM structures of DMC1 presynaptic and postsynaptic filaments.
We first solved the cryo-EM structures of human DMC1 filaments with ssDNA and dsDNA (homologous) in presynaptic and postsynaptic states (Structure 1, 3.33 Å and Structure 2, 3.47 Å, respectively), in the presence of adenylylimidodiphosphate (AMP-PNP) and $Ca^{2+}$. The detailed procedures of these two and three other structures addressed later are described in "Methods," whereas the workflows and original images and maps are shown in Supplementary Fig. 1 for all five filament structures. The parameters related to helical reconstructions and the detailed data collection, refinement, and validation statistics are listed in Supplementary Table 1. Similar to the corresponding complexes of RAD51[28,35], both structures adopt a well-ordered helical structure, with six protein protomers interacting with six triplets of DNA (Supplementary Fig. 2).

Detailed structures of these two DMC1 filaments and their comparisons with the corresponding human RAD51 complexes[28] and other recombinases are shown in Supplementary Fig. 3 (protomer interfaces) and Supplementary Fig. 4 (ATP-binding site). Here we focus on the identification of factors responsible for the mismatch tolerance (low fidelity) of DMC1 and the contrasting high fidelity of RAD51.

### Structure-based identification of fidelity-governing residues.
Recently, Greene and colleagues[17] reported that the difference between the fidelity of Saccharomyces cerevisiae (Sc) Dmc1 and Rad51 can be attributed to three lineage-specific residues of Loop 1, corresponding to human DMC1 E241/Q244/K245 and RAD51 A240/M243/H244, respectively, based on functional and biophysical analyses of a variety of variants. Figure 1b, c show the comparison between the structures of ssDNA in the presynaptic complexes of human DMC1 and RAD51, whereas Fig. 1d–f show the same comparison for dsDNA in the postsynaptic complexes. Interestingly, of the eight lineage-specific residues, only three (red asterisks in Fig. 1a) are located in the close proximity of DNA: Q244/P274/G275 of DMC1 (corresponding to M243/V273/D274 of RAD51) and they show different structural properties. The sidechain of Loop 1 residue Gln244 (polar) in DMC1 points toward the backbone of the dsDNA, whereas the corresponding Met243 (hydrophobic) in RAD51 points away (Fig. 1d–f). Furthermore, the Loop 2 residues in RAD51, V273 and D274 can form a V273:L238 hydrophobic/steric gate and a D274:R235 salt bridge, which can "lock up" the neighboring triplet basepairs. In DMC1, the hydrophobic/steric effect is smaller and the salt bridge is absent, which can "loosen up" the control for the triplet and allow some conformational flexibility for the basepairs, leading to mismatch tolerance and lower fidelity.

These analyses led us to propose a model (Fig. 1g) where Loop 1 residue Q244 and Loop 2 residues P274/G275 jointly contribute to the mismatch tolerance of DMC1 by stabilizing the DNA backbone and providing extra room and flexibility within the basepair triplet, to accommodate bulkier or staggered mismatched bases. In the following sections, we describe three additional structures and MD simulations to support these structural predictions, followed by functional validation.

### Structure of mismatched filaments from DMC1 but not RAD51.
Similar to DNA polymerases for which mismatched complexes are often too unstable to be solved, except for low-fidelity polymerases[36], we found that, with mismatched DNA, DMC1 forms well-ordered filaments, whereas RAD51 forms irregular filaments and two-dimensional (2D) class averages (Supplementary Fig. 5). We then successfully solved the structure of the mismatched dsDNA complex of DMC1 filaments (Structure 3, 3.36 Å). As the binding is not sequence-specific and as there could only be one mismatch in 15 basepairs (Oligo 3/5 with an A–C mismatch in Supplementary Table 2) for obtaining a stable mismatch structure, the densities of basepairs are averaged out in both homologous and mismatched structures. Nonetheless, the three-dimensional (3D) maps, including DNA backbone and their interacting residues, are essentially identical between the homologous (Structure 2) and mismatched (Structure 3) DMC1 postsynaptic filaments (Supplementary Fig. 6, with a root-mean-squared deviation (RMSD) of 0.53 Å out of a total of 309 Cα-atoms). These results provide a strong structural support for the mismatch tolerance of DMC1.

### Stabilization of DNA by lineage-specific Gln244 and conserved Arg242.
We next solved the structure of the postsynaptic filaments of DMC1-Q244M with homologous dsDNA (Structure 4,

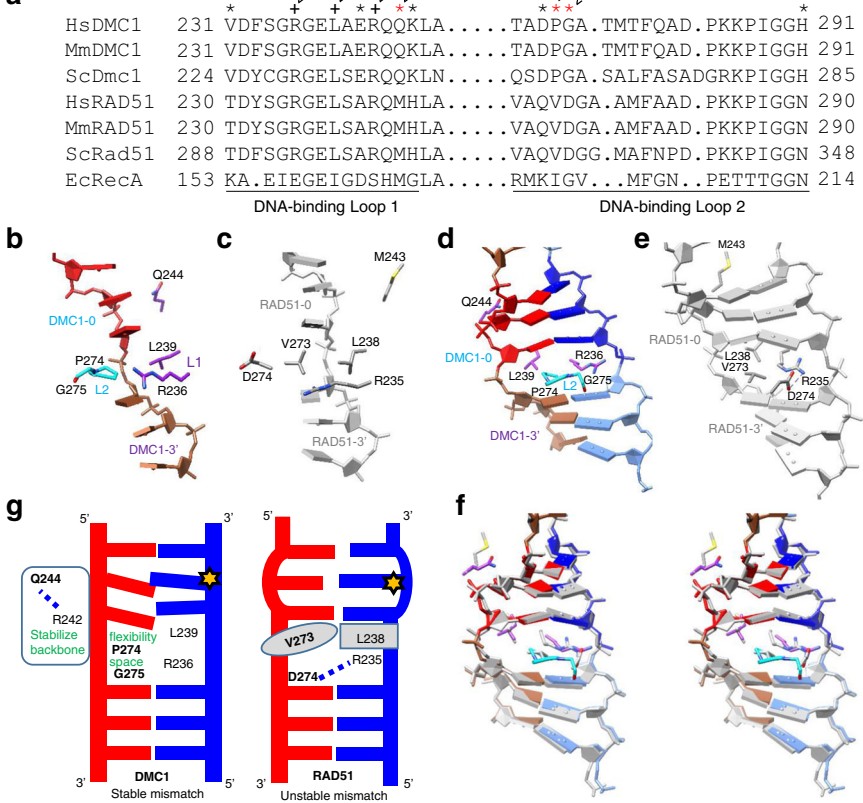

**Fig. 1 Possible fidelity-controlling residues of DMC1 and RAD51. a** Alignment of the amino acid residues in the Loop 1 and Loop 2 of DNA recombinases from *H. sapiens* (Hs), *M. musculus* (Mm), *S. cerevisiae* (Sc), and *E. coli* (Ec). All asterisks denote Loop 1 and Loop 2 lineage-specific amino acids, whereas red asterisks denote the three key residues identified in this study. Three conserved residues also addressed in this study are indicated by a + sign. **b**, **c** Inter-triplet area of DMC1 and RAD51 (PDB 5H1B[28]) in the presynaptic filament. **d**, **e** The same as **b**, **c**, except for the postsynaptic filaments of DMC1 and RAD51 (PDB 5H1C[28]). **f** Stereo view of the superposed **d** and **e**. The conserved residues DMC1 R236 and L239 from Loop 1, and the lineage-specific residues DMC1 P274 and G275 from Loop 2 are shown in sticks. The labels of DMC1 protomers are based on the designation of RAD51 protomers interacting with a nucleotide triplet from 5′ to 3′ as RAD51-5′, RAD51-0, and RAD51-3′ as described in Xu et al.[28]. **g** A schematic model showing the three lineage-specific residues (bolded) coordinating with three conserved residues (non-bolded) to contribute to the mismatch tolerance (low fidelity) of DMC1 and the unstable mismatch (high fidelity) of RAD51.

3.20 Å). Interestingly, the central axis of 2D class averages, where the DNA is located, is different between DMC1 and DMC1-Q244M (Fig. 2a), and only minor residual DNA maps were detected in the 3D maps of the latter (Fig. 2b). The results suggest that the Q244M mutant still binds DNA for filament formation but the bound DNA is too flexible to be observed by cryo-EM, likely due to the unfavorable replacement of a polar glutamine with a hydrophobic methionine. On the other hand, the protein structure of Q244M filaments is well resolved and is nearly identical to the structure of wild-type (WT) filaments (the backbone RMSD is 0.80 Å out of 305 $C_\alpha$ atoms), except that the Met244 sidechain points away from the DNA position (Fig. 2c) similar to that in RAD51 (Fig. 1e). The mismatched dsDNA was also unobservable in its complex with Q244M, but the protein structure could not be solved to a good resolution.

Importantly, as shown in Fig. 2d, the lineage-specific Gln244 appears to coordinate with the conserved Arg242 to stabilize the backbone of DNA in DMC1, with their sidechains spanning three basepairs and reaching out to two non-adjacent phosphodiester moieties. Even though the sidechain of Gln244 in this duet is somewhat far from the phosphodiester (4.4 Å for homologous and 4.2 Å for mismatched DNA) for a direct interaction, its functional role in stabilizing DNA backbone cannot be ruled out considering the moderate resolution of the cryo-EM structures and the change of sidechain orientation in the DMC1-Q244M

mutant and WT RAD51. Furthermore, due to backbone interactions between Gln244 and Arg242, the sidechain guanidinium group of Arg242 turns away from DNA in Q244M (3.1 Å–4.4 Å, Fig. 2d). These results suggest that the lineage-specific Gln244 coordinates with the conserved Arg242 to stabilize the backbone of DNA in DMC1, which is weakened in DMC1-Q244M, leading to enhanced DNA dynamics and loss of DNA density.

In comparison, such stabilization force is weaker in WT RAD51, with its hydrophobic Met243 pointing away from phosphodiester and Arg242 sidechain 3.5 Å from phosphate (PDB code 5H1C[28]). With the weakened backbone interaction, RAD51 is less able to stabilize mismatched DNA, as further supported by MD simulations described later.

**Increased solvent-accessible volume by Loop 2 P274/G275.** To test the hypothesis that DMC1 Loop 2 residues P274/G275 (relative to V273/D274 in RAD51) provide flexibility for DNA basepairs, we further solved the structure of the dsDNA filaments of the double mutant RAD51-VpDg (where the small letters represent corresponding residues from DMC1) (Structure 5, 3.43 Å). Comparison of the solvent-accessible volume of the cavity (Fig. 3a) shows that the volume around D274 and R235 of RAD51 is cleaved by the salt-bridge interaction, suggesting a rigid environment. In the double mutant RAD51-VpDg, the volume

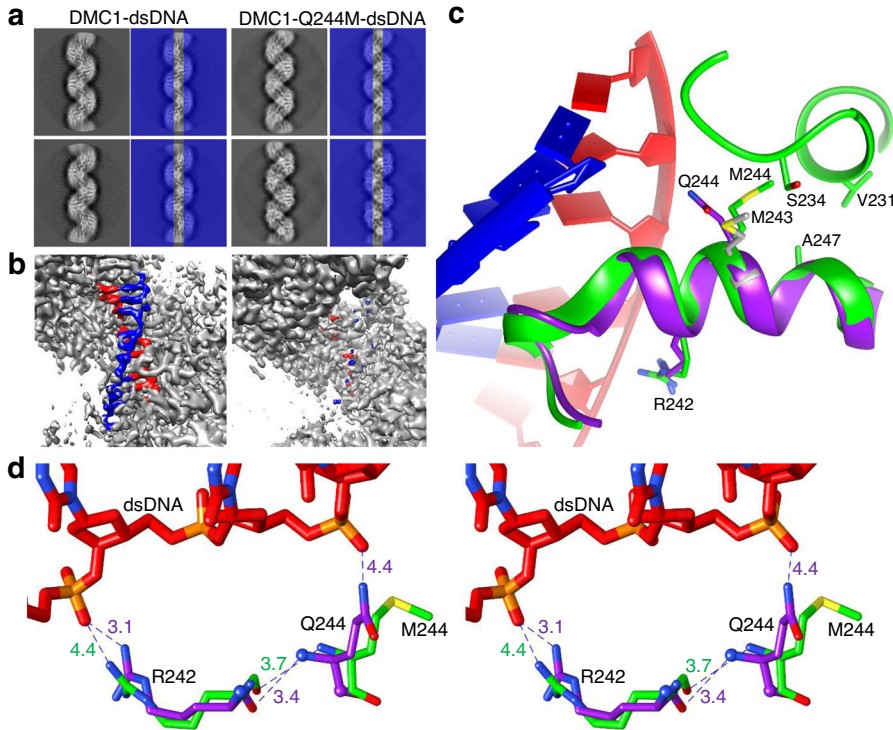

**Fig. 2 Structural comparison between the postsynaptic filaments of DMC1 and DMC1-Q244M with homologous dsDNA. a** Examples of 2D class average showing DNA in the central axis (highlighted by covering the two sides by blue rectangles with 50% transparency) of the DMC1 filaments but not the DMC1-Q244M filaments. **b** Three-dimensional cryo-EM density maps showing that DNA densities (colored in red and blue) can be clearly observed in the DMC1-WT filaments but only residual density can be detected in the DMC1-Q244M filaments. **c** Superposition of the two structures, showing that the polar sidechain of Gln244 in DMC1 (purple) points toward DNA, whereas that of hydrophobic Met244 in DMC1-Q244M (green) points away from DNA and toward CH groups of nearby residues. For comparison, Met243 (gray) in RAD51 postsynaptic complex is also shown. **d** Stereo view of the relationship between the sidechains of the Gln244-Arg242 pair and the backbone of dsDNA in the postsynaptic filaments of DMC1-WT (purple) and DMC1-Q244M (green). Relevant interatomic distances are shown in Å.

increases again, similar to DMC1. These differences can lead to extra space or flexibility of the triplet boundary to allow for mismatch tolerance by DMC1.

**Stabilization of both basepair and backbone by DMC1**. We then further performed MD simulations to compare homologous and mismatched structures for the three postsynaptic filaments where the structure with bound dsDNA was available to provide starting coordinates: DMC1, RAD51, and RAD51-VpDg. The RMSD values of the position differences of DNA were calculated throughout the MD simulations to evaluate the mismatch tolerability of each protein, based on their RMSD distributions (Supplementary Fig. 7). As shown in Fig. 3b, the RMSD values of mismatched DNA are substantially larger than those of homologous DNA for RAD51, but only slightly different for DMC1 and RAD51-VpDg, supporting that DMC1 can stabilize mismatched basepairs. Furthermore, plots of the RMSD distributions with DNA backbone only (Fig. 3c) show that the DNA backbone contributes substantially to the RMSD divergence for the mismatched filaments of RAD51 as indicated by the red arrow. In support, the structural ensembles in Fig. 3d show that in the RAD51-mismatch filaments, the DNA backbone, not just the basepairs, is significantly perturbed as highlighted by dashed boxes. The results taken together support that DMC1 can tolerate mismatched dsDNA by stabilizing not only the basepairs (by providing space for misalignment, Fig. 3a) but also the DNA backbone, as proposed in the structural model (Fig. 1g).

**Functional activities of DMC1 and RAD51 variants**. To validate the roles of Q244, P274, and G275 in the mismatch tolerability of DMC1 based on structural analyses, we performed two functional assays for DMC1, RAD51, and three reciprocal mutants of each: DMC1-Qm, RAD51-Mq, DMC1-PvGd, RAD51-VpGd, DMC1-QmPvGd, and RAD51-MqVpDg. For the fidelity comparison to be meaningful, we have ensured that the DNA-binding and strand-exchange activities of the mutants were not substantially perturbed (Supplementary Fig. 8), with the exception of DMC1-QmPvGd, which was therefore excluded. The results of all other mutants are summarized in Fig. 4.

First, the mismatch tolerability was examined by a fluorescence-based DNA strand-exchange assay (Fig. 4a, b)[37]. The presynaptic filament was assembled on a ssDNA with a sequence for homologous or mismatched strand exchange. Then, the Cy5 and Cy3 double-labeled dsDNA was added to initiate the strand-exchange reaction and the fluorescence signal of Cy3 was monitored in real-time, which increases as the Cy3-labeled ssDNA is released from the three-strand intermediate, because Cy3 is no longer close to Cy5. As shown by the time course plots (Fig. 4c) and the bar plots of % mismatch tolerance (Fig. 4d), compared to the mismatch tolerance of WT human DMC1 at 99.4%, WT human RAD51 displayed 67%, comparable to its *Escherichia coli* ortholog RecA (65.4%, also reported previously[29]). The Loop 1 reciprocal mutants DMC1-Qm and RAD51-Mq changed in opposite directions to 77% and 87%, respectively. Analogously, the Loop 2 reciprocal mutants DMC1-PvGd and RAD51-VpDg changed to 81% and 100%, respectively. As expected, the triple

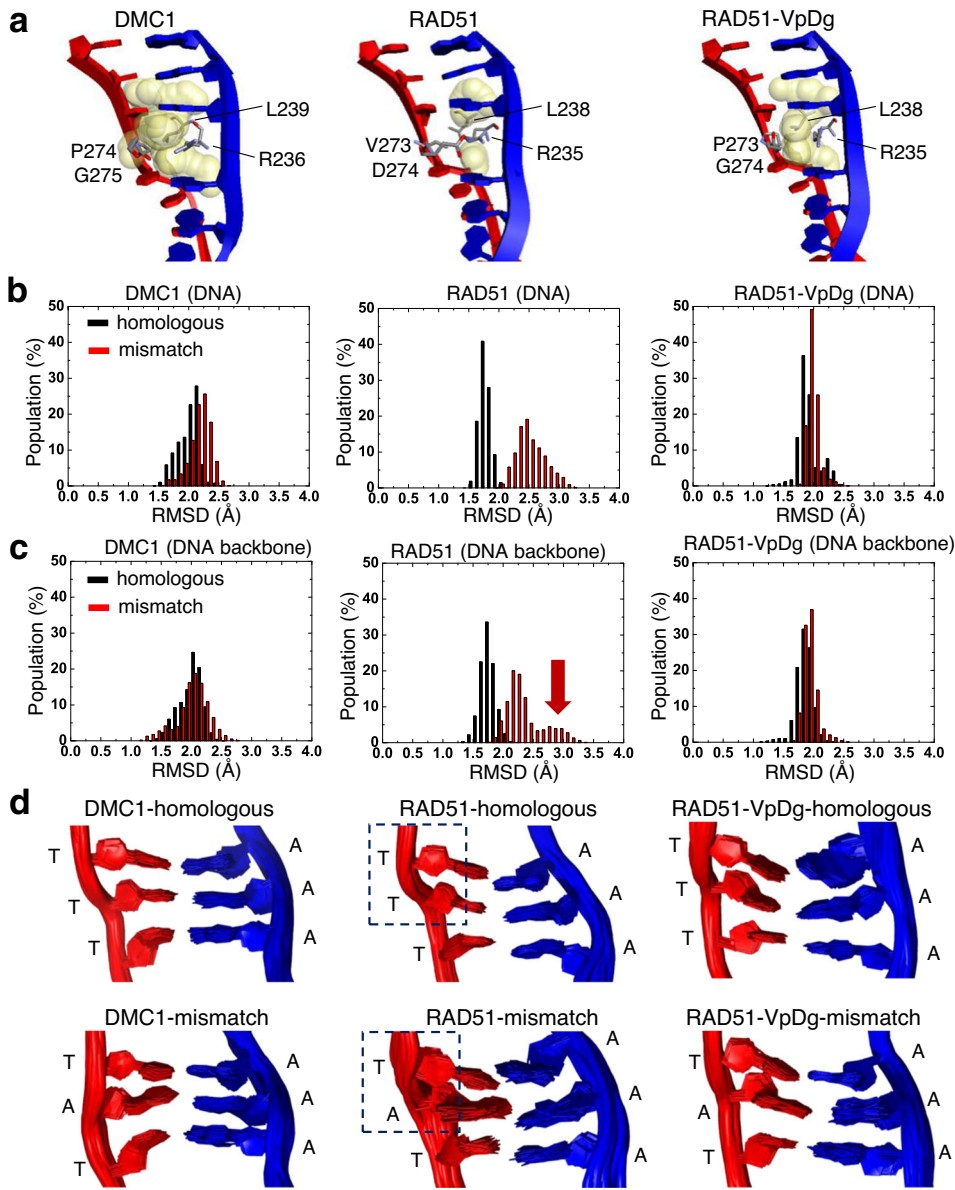

**Fig. 3 Flexibility of the basepair triplet in the postsynaptic filaments of DMC1, RAD51, and RAD51-VpDg. a** Solvent accessibility within 5 Å around the four gate-keeping residues. The cavities were estimated by point atoms, with radii equal to 1.4 Å. The cavity is colored by yellow and the residues are shown as gray stick. **b**, **c** RMSD distributions for homologous and mismatched DNA from MD simulations, plotted for all DNA atoms (9 basepairs, 574 atoms including hydrogen) (**b**) and DNA backbone only (106 atoms) (**c**). The arrow indicates the contribution of DNA backbone to the high-RMSD structures. The overall RMSD fluctuations are shown in Supplementary Fig. 7. **d** Structural ensembles of the three middle basepairs from MD simulations. The dashed boxes highlight the changes in the DNA backbone between homologous and mismatched RAD51 filaments.

mutant RAD51-MqVpDg also gained 100% mismatch tolerance activity of DMC1.

We next used single-molecule fluorescence colocalization experiments to monitor the stability of the recombinase-mediated three-strand intermediate, the triplex state (Fig. 4e). Surface-anchored and Cy5-labeled DNA substrates containing 81 nt ssDNA were incubated with the recombinase and paired with Cy3-labeled duplex DNA with 15 nt homology, 15 nt containing one mismatch, or 12 nt homology. By monitoring the amount of colocalized Cy3/Cy5 signal at a given time, we can determine the lifetime of the triplex state and thus the dissociation rate (see plots and values in Fig. 4f). In all panels, the 12 nt homology sample showed the largest dissociation rate, because the short continuous stretch reduces the stability of the triplex state. As expected, although the 15 nt mismatch substrate is closer to the 15 nt

homologous substrate than the 12 nt homologous one in WT DMC1, it is closer to the 12 nt homo in WT RAD51. As shown in the other panels in Fig. 4f, both DMC1-Qm and DMC1-PvGd mutants displayed similar properties to WT RAD51, whereas the reciprocal RAD51 mutants as well as the triple mutant RAD51-MqVpDg behaved similar to WT DMC1.

The results from the two different functional assays taken together support that the Loop 1 residue Q244 and the two Loop 2 residues P274/G275 of DMC1, and the corresponding M243/V273/D274 of RAD51, play key roles in governing the fidelity of both recombinases.

Overall, analyses of five filament structures and MD simulations of WT and mutant recombinases support a model in which DMC1 Loop 2 P274/G275 loosen the gate between basepair triplets to provide space and flexibility for mismatched basepairing, whereas

Loop 1 Gln244 coordinates with Arg242 to help stabilize the DNA backbone. Although not described in detail, several other conserved residues in Loops 1 and 2 are also intimately involved in DNA binding indirectly, but the three lineage-specific residues addressed here play key roles in the structural differences between DMC1 and RAD51 filaments. Functional analyses show that mutants of either Loop 1 or Loop 2 can change the fidelity of one recombinase more than half way to the other and the triple mutant RAD51-MqVpDg is as mismatch-tolerant as DMC1, validating the predictions from structural analyses.

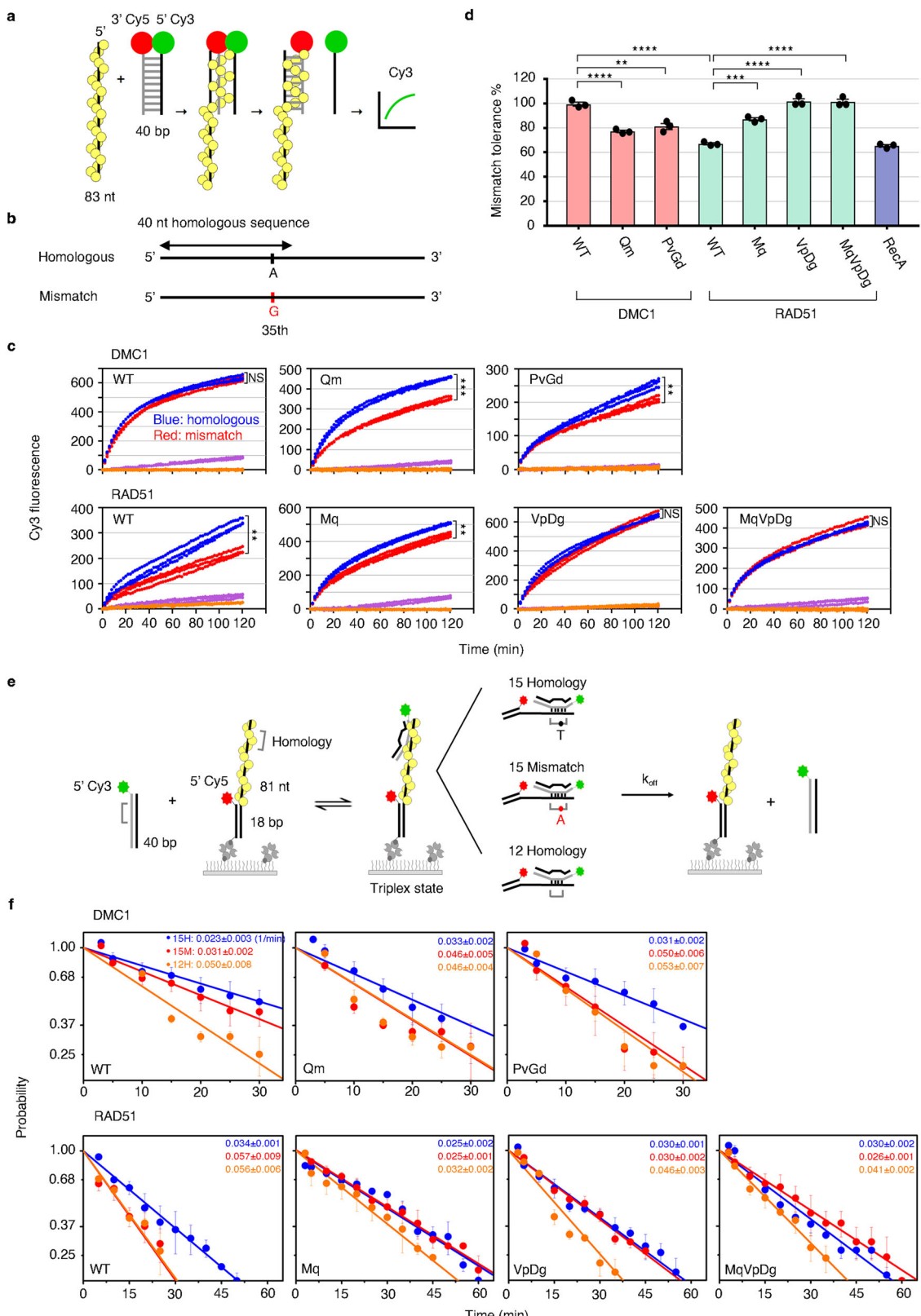

**Fig. 4 Functional analysis of the mismatch tolerability of DMC1 and RAD51 variants for Loop 1 and Loop 2 residues. a** Schematics of fluorescence-based DNA strand exchange. Yellow balls represent the recombinase from filaments on ssDNA. Red and green balls represent Cy5 and Cy3 fluorophores, respectively, at the end of donor dsDNA. **b** Design diagram of homologous and mismatched substrates. **c** Cy3 fluorescence signals generated by DMC1, RAD51, and their mutant variants-mediated strand-exchange activity with homologous or mismatch substrates were monitored in real time. Blue and red dots represent the Cy3 fluorescence spectra from the homologous and mismatch substrates, respectively, while purple and orange dots represent the nonhomologous substrate with ATP and homologous substrate without ATP, respectively. Data shown are from three independent experiments. Statistics was performed by two-sides repeated-measure ANOVA (NS, not significant, $**P < 0.01$ and $***P < 0.001$). The $P$-values of DMC1-WT, Qm, and PvGd are 0.0757, <0.001, and 0.005, respectively, whereas for RAD51-WT, Mq, VpDg, and MqVpDg are 0.007, 0.003, 0.224, and 0.806, respectively. **d** Bar plots for the percentage of mismatch tolerance determined by the final fluorescence intensity of mismatch substrate relative to homologous substrate at 120 min. Data shown are average values (mean) ± SEM from three independent experiments ($n = 3$). Statistics were performed by one-way ANOVA with Tukey's post hoc test ($**P < 0.01$, $***P < 0.001$, $****P < 0.0001$). The $P$-values of DMC1-WT vs. PvGd and RAD51-WT vs. Mq are 0.0012 and 0.0003, respectively, whereas that of the rest are <0.0001. **e** Experimental design of single-molecule fluorescence colocalization to determine the stability of the recombinase-mediated triplex state. **f** Dissociation of the triplex state as a function of time upon removal of excess duplex strand. Three different DNA substrates (15 nt homo, 15 nt mismatch, and 12 nt homo) were determined. Slopes of the plots are the dissociation rates (min$^{-1}$), which are shown within each panel by blue, red, and orange texts for the 15 nt homo (15H), 15 nt mismatch (15M), and 12 nt homo (12H), respectively. Data (mean ± SEM) are from at least three independent experiments ($n = 3$, except for DMC1-PvGd-12H, $n = 4$; RAD51-WT-15M, and 12H, $n = 5$). Raw data are provided in the Source Data file.

## Methods

**Plasmids**. The plasmid harboring human *DMC1* cDNA in vector pRSFDuet (Novagen)[38] was subjected to site-directed mutagenesis to construct the DMC1 mutant variants Q244M, PvGd and QmPvGd. The plasmid harboring human *RAD51* cDNA in vector pET11 (Novagen)[39] was subjected to site-directed mutagenesis to construct the RAD51 mutant variants M243Q, VpDg, and MqVpDg. All the resulting plasmids were sequenced to ensure there is no unwanted mutation.

**DNA substrates**. All oligonucleotides with their sequences are listed in Supplementary Table 2. All oligonucleotides were synthesized and gel purified by Genomics BioSci & Tech. All of the duplex DNA substrates were prepared by incubating two complementary oligonucleotides in the annealing buffer (50 mM Tris-HCl pH 7.5, 10 mM MgCl₂, 100 mM NaCl, and 1 mM dithiothreitol (DTT)) at 80 °C for 3 min, then at 65 °C for 3 min, followed by slow cooling to 23 °C for DNA annealing. The resulting duplex was purified from a 10% polyacrylamide gel by electro-elution and filter-dialyzed into TE buffer (10 mM Tris-HCl pH 8.0 and 0.5 mM EDTA) by a Centricon-10 concentrator (Millipore) at 4 °C.

For cryo-EM, the 80-mer Oligo 1 was used for the assembling of presynaptic filament. The duplex DNA annealed by 15-mer Oligo 3 and Oligo 4 (homologous) or Oligo 3 and Oligo 5 (mismatched) was used for the assembling of postsynaptic filament.

For DNA mobility shift analysis, the Oligo 1 was 5′-end-labeled with [γ-³²P] ATP (PerkinElmer) by polynucleotide kinase (New England Biolabs). Following removal of the unincorporated nucleotide with a Spin 6 column (Bio-Rad), the radiolabeled Oligo 1 was used as the ssDNA substrate. The duplex DNA annealed by radiolabeled Oligo 1 and its exact complement Oligo 2 was used as the dsDNA substrate. For isotope-labeled DNA strand-exchange reactions, the ssDNA Oligo 1 was used to assemble nucleoprotein filament and the 40 bp duplex annealed by Oligo 6 and radiolabeled Oligo 7 was used as the homologous donor dsDNA. For fluorescence-based DNA strand exchange, the 83-mer Oligo 8 (homologous ssDNA), Oligo 9 (mismatched ssDNA), and Oligo 1 (nonhomologous ssDNA) were used to assemble nucleoprotein filament, and the 40 bp duplex annealed by Oligo 10 with 3′-Cy5 fluorophore labeled and Oligo 11 with 5′-Cy3 fluorophore labeled was used as the donor dsDNA.

For single-molecule triplex-state stability experiments, the 3′-biotinylated Cy5-labeled hybrid DNA was used for the assembling of presynaptic filament. To prepare the hybrid DNA, 18-mer 3′-biotinylated Oligo 12 with 5′-Cy5 fluorophore labeled was annealed with three different 99-mer ssDNA, respectively: Oligo 13 containing 15 nt full homology, Oligo 14 containing 15 nt complementary sequence with one mismatch, and Oligo 15 containing 12 nt full homology. Cy3-labeled 40 bp dsDNA annealed by Oligo 16 and Oligo 17 was used as the donor dsDNA to form the triplex state.

**Protein expression and purification**. His₆-tagged WT human DMC1 was overexpressed in the *E. coli* RecA-deficient BLR strain harboring with pRARE to supply tRNAs for rare codons. For purification of the DMC1 protein[38], the clarified cell lysate was subjected to TALON affinity resin (TaKaRa), Source Q column (GE Healthcare), macrohydroxyapatite column (GE Healthcare), and Mono Q 5/50 GL column (GE Healthcare). The DMC1-containing fractions were pooled, concentrated in a Centricon-30 concentrator (Millipore), and stored in small aliquots at −80 °C. The DMC1 variants were expressed and purified as described for the WT protein.

Human RAD51 was expressed in *E. coli* BLR pRARE strain. The protein purification steps were modified from previously described procedure[40]. Briefly, cell extract was subjected to ammonium sulfate (40% saturation) precipitation and

resuspended in buffer A (20 mM K₂HPO₄ pH 7.5, 0.5 mM EDTA, 10% glycerol, 0.01% Igepal, 1 mM 2-mercaptoethanol) supplemented with 2 mM Benzamidine, 1 mM phenylmethylsulfonyl fluoride, and 3 μg/ml of the following protease inhibitors: Aprotinin, Chymostatin, Leupeptin, and Pepstatin A. The RAD51 suspension was then chromatographically fractionated in a Sepharose Q column by applying a 180 ml gradient of 150~660 mM KCl in buffer A. The RAD51 pool was diluted with buffer A and further fractionated in a 1 ml macrohydroxyapatite column using a 90 ml linear gradient of 70~560 mM KH₂PO₄ in buffer A containing 50 mM KCl. The pooled RAD51-containing fraction was diluted and further purified by a 1 ml Source Q column with an 80 ml 235~575 mM KCl gradient in buffer A, and following by a 1 ml Mono Q column with a 45 ml 235~490 mM KCl gradient in buffer A. Finally, the RAD51-containing fractions were pooled, concentrated, and stored at −80 °C. The RAD51 variants were expressed and purified as described for the WT protein. *E. coli* RecA protein was purchased from New England Biolabs.

**Cryo-EM sample preparation and data acquisition**. Both presynaptic and postsynaptic complex assembly reaction were incubated with 4 μM DMC1 (or RAD51) protein and 24 μM nucleotides in buffer B (35 mM Tris-HCl pH 7.5, 108 mM KCl and 1 mM DTT) containing 2 mM AMP-PNP and 5 mM CaCl₂ at 37 °C for 30 min. Four microliters of protein sample were applied onto a pre-glow-discharged graphene-oxide-coated Quantifoil holey carbon grid (1.2/1.3, 200 mesh) using published protocol[41]. The grids were blotted for 1 s at 22 °C with 100% relative humidity and plunge-frozen in liquid ethane cooled by liquid nitrogen using a Vitrobot Mark IV (Thermo Fisher). Cryo-EM data were acquired on a Titan Krios G3 (Thermo Fisher) microscope operated at 300 keV, equipped with a GIF Quantum K2 detector system (Gatan). Automated data acquisition was carried out using EPU software (Thermo Fisher) at a nominal magnification of ×165,000, yielding a pixel size of 0.84 Å. Movies of 50 frames, corresponding to a total dose of 50 e⁻ Å⁻² were collected in counting mode at a dose rate of 1.0 e⁻ Å⁻² per frame.

**Image processing and helical reconstruction**. The procedures are summarized in Supplementary Fig. 1. All movies were motion-corrected and dose-weighted using RELION's own implementation[42]. Only aligned, non-dose-weighted micrographs were then used to estimate the contrast transfer function with Gctf[43]. All subsequent image-processing steps were carried out using helical reconstruction in RELION 3.0[44]. Filaments were manually picked and segments were extracted using a box size of 384 pixels and an inter-box distance of ~10% of the box length. A spherical mask with a diameter equal to 90% of the box size was applied. Iterative rounds of 2D classification were performed to remove low-quality filaments. A simple cylinder was used as initial model to prevent model bias. Together with the selected particles, 3D classification was performed using the helical rise and twist derived from RAD51 presynaptic filaments as initial helical parameters[28,35]. The reconstruction served as an initial reference model for 3D auto-refinement. All refinements were carried out following the gold-standard procedure where the data set was divided into two half-sets. After refinement was converged, the corrected Fourier shell correlation (0.143 Å) criterion was calculated to estimate the resolution after post-processing with a soft mask applied.

**Model building and refinement**. The initial atomic model of human DMC1 protomers in both the presynaptic and postsynaptic complexes was generated from homology modeling of the cryo-EM structure of human RAD51 filaments[28,45]. The structures were then docked into the EM map by using PHENIX[46]. DNA and AMP-PNP were manually appended in COOT[47] and rigid-body docked into the electron density map in UCSF-Chimera[48]. The built models were replicated

multiple times and fitted into the EM maps of the presynaptic and postsynaptic complexes in UCSF-Chimera by following their corresponding helical symmetries to generate the atomic models of longer assemblies. The models of presynaptic and postsynaptic complexes were further refined in Rosetta[49]. The refined atomic models were further validated and manually inspected using PHENIX[46] and COOT[47]. Since the binding is not sequence-specific and the densities for the bases are averages, they were fitted with poly-dT (ssDNA) and poly-dT:poly-dA (dsDNA, invading strand and complementary strand, respectively) for model building.

**MD simulations**. MD simulations in explicit water were conducted using the Amber 16 package[50]. For DMC1 and RAD51-VpDg, our cryo-EM models (consisting of three protein protomers and nine A-T basepairs) provided the starting protein coordinates. For RAD51, the initial structure was based on PDB 5H1C[28]. All single mismatch-containing DNA was rebuilt by Web 3DNA 2.0[51], where the Thymine in the middle position was mutated to Adenine. The simulations were performed using Amber ff14SB force field[52] and the residue charges were calculated based on the libraries in the Amber 16 package. Periodic boundary conditions were imposed with box lengths of $107.5 \times 95.1 \times 121.8$ Å$^3$. The atoms of some loop 1 and loop 2 residues, e.g., R236, P274, G275, and Q244 of DMC1 were constrained. A Langevin thermostat was used to maintain the system temperature with collision frequency of $1$ ps$^{-1}$ to the target temperature 300 K. The SHAKE algorithm was implemented to constrain the covalent bond. All MD simulations were carried out in the isothermal–isobaric ensemble with the Langevin thermostat and the time step of 1 fs. The system first underwent an annealing process from 0 to 300 K under a constant pressure of 1.0 bar over 7 ns and maintained at this equilibrated point for the following simulation. After equilibrated steps, the final equilibrium system density was ~$1.0 \pm 0.01$ g/cm$^3$. Finally, we further performed 13 ns MD simulations for checking systems equilibrium and RMSD analysis.

**Electrophoretic mobility shift assay**. The $^{32}$P-labeled 80-mer ssDNA (3 μM nucleotides) and $^{32}$P-labeled 80 bp dsDNA (3 μM basepairs) were incubated individually with the indicated amount of human DMC1 or RAD51 variants in 10 μl of buffer C (35 mM Tris-HCl pH 7.5, 1 mM DTT, 10 mM MgCl$_2$, 100 ng/μl bovine serum albumin (BSA), 50 mM KCl and 2 mM ATP) at 37 °C for 10 min. The reaction mixtures were fractionated in 2% agarose gels in TAE buffer (40 mM Tris, 20 mM acetate pH 7.5, and 2 mM EDTA) at 4 °C. The gels were dried onto DE81 paper and subjected to phosphorimaging analysis (Bio-Rad Personal Molecular Imager) and quantitative analysis (Bio-Rad Quantity One 4.6.9).

**DNA strand-exchange assay**. The 80-mer Oligo 1 (4.8 μM nucleotides) was pre-incubated with the indicated amount of human DMC1 or RAD51 variants individually to assemble nucleoprotein filament in buffer C supplemented with 10 mM CaCl$_2$ at 37 °C for 5 min. The reaction was initiated by adding homologous $^{32}$P-labeled 40-mer duplex (2.4 μM basepairs) to a final volume of 12.5 μl. After a 30 min incubation, a 5 μl aliquot was removed, mixed with an equal volume of 0.1% SDS containing proteinase K (1 mg/ml), and incubated at 37 °C for 15 min. The samples were fractionated in 10% polyacrylamide gels in TBE buffer (89 mM Tris, 89 mM borate, and 2 mM EDTA pH 8.0). Gels were dried onto DE81 paper and subjected to phosphorimaging and quantitative analysis.

**Fluorescence-based DNA strand exchange**. The 83-mer Oligo 8, Oligo 9, or 80-mer Oligo 1 (3 μM nucleotides) harboring the homologous, one mismatched, or nonhomologous sequence, respectively, was incubated with human DMC1 variants (1 μM for WT and Qm mutant, and 2 μM for PvGd mutant) or RAD51 variants (0.75 μM) to assemble nucleoprotein filament in 40 μl buffer C supplemented with 10 mM CaCl$_2$. After 30 min incubation on ice, 10 μl Cy5 and Cy3 double-labeled dsDNA (1.45 μM basepairs) were added to initiate the strand-exchange reaction. The reaction mixture was transferred into Falcon 384-Well Optilux Black/Clear Flat and the fluorescence emission of Cy3 at 590 nm (bandwidth: 35 nm) upon excitation at 530 nm (bandwidth: 25 nm) was monitored in real-time by Gene5 2.07 spectrofluorometer (BioTek SynergyHTX multi-mode reader). Data were collected every two min at room temperature. The Cy3 emission signal ($F_t$) representing the change in the amount of strand exchange product at each time point was calculated using following equation: $F_t = (S_t - S_0) - (B_t - B_0)$. "$S_t$" and "$S_0$" are the fluorescence intensity from the reaction containing protein at each time point or beginning, respectively, whereas "$B_t$" and "$B_0$" are from the reaction without protein. The fluorescence spectra of Cy3 signals were plotted by the time correlated to $F_t$. The relative strand-exchange efficiency ($R_f$ shown in %) for the calculation of mismatch tolerability of recombinase is calculated with the following equation: $R_f = F_M/F_H$, where "$F_M$" is the Cy3 emission signal ($F_t$) at the 120 min time point of the reaction with mismatched substrates and "$F_H$" is the Cy3 emission signal of the reaction with homologous substrates at the 120 min time point.

**Single-molecule triplex-state stability experiments**. The reaction chamber was constructed by incubating 0.02 mg/mL streptavidin for 5 min in the PEGylated slide and coverslip[53]. After washing excess streptavidin with buffer D (20 mM Tris-HCl and 50 mM NaCl), three different 3′-biotinylated Cy5-labeled hybrid DNA

containing 15 nt full homology, 15 nt with one mismatch, and 12nt full homology, individually, were immobilized on the surface for 5 min. It was then washed with buffer E (buffer F: 35 mM Tris-HCl pH 7.5, 1 mM DTT, 10 mM MgCl$_2$, 1 μg/μl BSA, 50 mM KCl, and 10 mM CaCl$_2$; supplementing with 4 mg/ml glucose, 30 U/ml glucose oxidase, and 30 U/ml catalase). Recombinase (1 μM RAD51/mutant or 2 μM DMC1/mutant) in buffer F supplemented with 2 mM ATP was then incubated with surface-bound hybrid DNA for 15 min to form nucleoprotein filaments. Triplex-state formation was carried out by washing excess recombinases and flowing 10 nM Cy3-labeled 40 bp dsDNA into the reaction chamber for 20 min. Free Cy3-labeled 40 bp dsDNA was then washed three times using imaging buffer E with 2 mM ATP but without CaCl$_2$ and it was defined as time zero. The numbers of triplex state were monitored at given time points by scoring the numbers of colocalized Cy3/Cy5 spots in the same field-of-view. At each time point, 20 frames (2 s) were recorded sequentially with green and red lasers to image Cy3 and Cy5 dyes, respectively.

We utilized objective-type total internal reflection fluorescence microscope (IX-71; Olympus) equipped with 532 and 633 nm lasers to alternatively excite Cy3 and Cy5. The fluorescence signal was acquired by an electron-multiplying CCD (ProEM 512B; Princeton Instruments) at 10 Hz using a dual-view system. The image recorded using a software program is written in LABVIEW 8.6. Data analysis was performed using MATLAB for automatic spot detection and colocalization. The software used for fitting was Origin.

**Statistical analysis**. The difference between experimental curve with homologous and mismatched substrates for each protein was analyzed with two-sides repeated-measures analysis of variance (ANOVA). Between-subject effects were determined by fluorescent data extracted from every 10 min and calculated with Statistical Product and Service Solutions. $P$-value $< 0.05$ indicates that the reaction is significantly different between homologous and mismatched substrates. The significance between mismatch tolerance of WT and mutant variant protein was analyzed with GraphPad Prism 7. Multiple groups were compared using one-way ANOVA with Tukey's post hoc test. $P$-values are represented as $P > 0.05$ is not significant and $P < 0.05$ is significant. The normality of data was tested by Shapiro–Wilk normality test to confirm that the data was normally distributed.

**Reporting summary**. Further information on the research design can be found in the Nature Research Reporting Summary linked to this paper.

## Data availability

The refined coordinates and corresponding cryo-EM maps have been deposited in the Protein Data Bank and the Electron Microscopy Data Bank under accession codes "PDB-7C9C" and EMD-30311 (DMC1 presynaptic complex), "PDB-7C98" and EMD-30308 (DMC1 postsynaptic complex), "PDB-7C99" and EMD-30309 (DMC1 postsynaptic complex with mismatch), "PDB-7CGY" and EMD-30366 (DMC1-Q244M mutant postsynaptic complex), "PDB-7C9A" and EMD-30310 (RAD51-V273P, D274G mutant postsynaptic complex). All other relevant data are described in the Supplementary Information. Any additional data related to this paper are available upon request. Source data are provided with this paper.

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

## Acknowledgements

This work was supported by Academia Sinica (M.-D.T., P.C., and M.-C.H.), National Taiwan University (P.C. and H.W.L.), Taiwan Ministry of Science and Technology (MOST 105-2314-B-002-073-MY4 and MOST 108-2321-B-002-054 to P.C., MOST-107-2113-M-002-010-MY3 to H.-W.L.), and Taiwan Protein Project (Grant number AS-KPQ-105-TPP and AS-KPQ-109-TPP2 to M.-D.T.). The cryo-EM experiments were performed at the Academia Sinica Cryo-EM Center (ASCEM) and the cryo-EM data were processed at the Academia Sinica Grid-computing Center (ASGC). ASCEM is supported by Academia Sinica (Grant number AS-CFII-108-110) and Taiwan Protein Project. ASGC is supported by Academia Sinica.

## Author contributions

P.C., M.-C.H., and M.-D.T. conceived and designed the research. S.-C.L., Y.-M.W., and M.-C.H. performed cryo-EM experiments and processed data. H.-Y.Y., H.-Y.C., G.-C.S., C.-Y.L., and P.C. prepared proteins and designed and performed functional assays. W.-H.L. and H.-W.L. designed and performed single-molecule assays. C.-H.Y. performed computational analysis. All authors analyzed data. M.-D.T., P.C., and M.-C.H. supervised the project and interpret overall data. S.-C.L., H.-Y.Y., W.-J.W., M.-C.H., P.C., and M.-D.T. wrote the paper.

## Competing interests

The authors declare no competing interests.
