## [Peer Review File · Nature Communications]

REVIEWERS' COMMENTS

Reviewer #1 (Remarks to the Author):

The authors have thoroughly addressed mine and the other reviewers' comments with significant additional data and discussion and the manuscript is much improved. Overall, the structural basis for the difference between Rad51 and Dmc1 with regard to mismatch tolerance is an important question, and the fact that the Rad51 triple mutant (MqVpDg) retains Dmc1-like mismatch tolerance is in strong agreement with the conclusion from the structures regarding the three key residues (out of eight that were possible). The additional structure of the Qm mutant of DMC1 provides additional support for the hypothesis that Gln244 restricts the backbone conformation. Overall the manuscript is significantly strengthened and suitable for publication.

Reviewer #2 (Remarks to the Author):

The authors have substantially improved their paper by adding new data and making changes to the manuscript. I do not have any further concerns.

Reviewer #3 (Remarks to the Author):

The authors have satisfactorily addressed all of my comments from the previous round of review. Indeed, they have added significant new data that greatly strengthen the manuscript. The authors are to be commended for this study and the work should be considered ready for publication.